# Foodomics-Based Approaches Shed Light on the Potential Protective Effects of Polyphenols in Inflammatory Bowel Disease

**DOI:** 10.3390/ijms241914619

**Published:** 2023-09-27

**Authors:** Giovanni Pratelli, Bartolo Tamburini, Daniela Carlisi, Anna De Blasio, Antonella D’Anneo, Sonia Emanuele, Antonietta Notaro, Federica Affranchi, Michela Giuliano, Aurelio Seidita, Marianna Lauricella, Diana Di Liberto

**Affiliations:** 1Department of Physics and Chemistry (DiFC) Emilio Segrè, University of Palermo, 90128 Palermo, Italy; giovanni.pratelli@unipa.it; 2Section of Biochemistry, Department of Biomedicine, Neurosciences and Advanced Diagnostics (BIND), University of Palermo, 90127 Palermo, Italy; bartolo.tamburini@unipa.it (B.T.); daniela.carlisi@unipa.it (D.C.); sonia.emanuele@unipa.it (S.E.); 3Department of Health Promotion, Mother and Child Care, Internal Medicine and Medical Specialties (ProMISE), University of Palermo, 90127 Palermo, Italy; aurelio.seidita@unipa.it; 4Laboratory of Biochemistry, Department of Biological, Chemical and Pharmaceutical Sciences and Technologies (STEBICEF), University of Palermo, 90127 Palermo, Italy; anna.deblasio@unipa.it (A.D.B.); antonella.danneo@unipa.it (A.D.); antonietta.notaro@unipa.it (A.N.); federica.affranchi@unipa.it (F.A.); michela.giuliano@unipa.it (M.G.)

**Keywords:** inflammatory bowel diseases, omics, polyphenols, inflammation, antioxidative properties, gut dysbiosis

## Abstract

Inflammatory bowel disease (IBD) is a chronic and progressive inflammatory disorder affecting the gastrointestinal tract (GT) caused by a wide range of genetic, microbial, and environmental factors. IBD is characterized by chronic inflammation and decreased gut microbial diversity, dysbiosis, with a lower number of beneficial bacteria and a concomitant increase in pathogenic species. It is well known that dysbiosis is closely related to the induction of inflammation and oxidative stress, the latter caused by an imbalance between reactive oxygen species (ROS) production and cellular antioxidant capacity, leading to cellular ROS accumulation. ROS are responsible for intestinal epithelium oxidative damage and the increased intestinal permeability found in IBD patients, and their reduction could represent a potential therapeutic strategy to limit IBD progression and alleviate its symptoms. Recent evidence has highlighted that dietary polyphenols, the natural antioxidants, can maintain redox equilibrium in the GT, preventing gut dysbiosis, intestinal epithelium damage, and radical inflammatory responses. Here, we suggest that the relatively new foodomics approaches, together with new technologies for promoting the antioxidative properties of dietary polyphenols, including novel delivery systems, chemical modifications, and combination strategies, may provide critical insights to determine the clinical value of polyphenols for IBD therapy and a comprehensive perspective for implementing natural antioxidants as potential IBD candidate treatment.

## 1. Introduction

The first important interactions of nutrients with the human body occur at the level of the gastrointestinal tract (GT), whose primary function is the digestion and absorption of food, including solids and liquids. Notably, GT also plays an important role as a barrier against many potentially harmful ingested substances. The epithelial barrier’s functionality is ensured by tight junctions, adherens junctions, and desmosomes. Moreover, epithelial cells fully contribute to the defense of the intestine by secreting bactericidal substances, such as defensins, both naturally and in response to the pattern recognition receptors (PRRs) against microbial pathogen-associated molecular patterns (PAMPs) [1]. Thus, the health and defense of the human body depend on the integrity of the gut mucosa, so that, for years now, many authors have identified a new condition: the “Leaky Gut Syndrome” [2].

The integrity of GT is also correlated with the presence of the so-called intestinal microbiota. The number of microorganisms inhabiting the gastrointestinal tract is estimated to exceed 10^14^ (100 trillion) [3], with a rich bacterial diversity containing more than 1000 species [4] and the most represented phyla are Firmicutes (49–76%), Bacteroidetes (16–23%), Proteobacteria, and Actinobacteria [5]. These bacteria are responsible for mineral absorption and carbohydrate degradation [6], as well as for amino acids and vitamins synthesis [7]. In addition, gut microbiota enhances mucosal immunity tolerance through metabolizing dietary fiber into short-chain fatty acids (SCFAs) [8], such as butyrate, propionate, acetate, and activating regulatory T cells (Tregs), a T cell subset specialized in suppressing the immune response and maintaining self-tolerance and tissue homeostasis [9,10].

Furthermore, the production of indoles and their derivatives, such as tryptophan metabolites and indoleacrylic acid, by intestinal commensal *Clostridium sporogenes* is responsible for the activation of the aryl hydrocarbon receptor, the induction of mucin family gene expression, and the activation of the nuclear factor erythroid 2-related factor 2 (Nrf2), a crucial regulator of the cellular defense mechanisms against xenobiotic and oxidative stress, thus enhancing the expression of antioxidant enzymes and molecules in the GT [11,12].

At the same time, Bacteroides species metabolize dietary fiber into succinate [13], which is responsible for the stabilization of hypoxia-inducible factor-1α (HIF-1α) and the activation of dendritic cells, thus promoting IL-1β production with the consequent triggering of an inflammatory response [14]. In addition, inflammatory responses are also modulated by sphingolipids, especially ceramide, synthesized by Bacteroides via bacterial serine-palmitoyl transferase, which turns off the inflammation in host cells [15,16]. Finally, the decrease in pH caused by the high concentration of SCFA prevents the growth of potentially pathogenic bacteria, such as *Escherichia coli* and other members of the Enterobacteriaceae [17]. Due to the close relationship between the intestinal barrier and microbiota, the gastrointestinal system functions as efficiently as possible, making a significant contribution to homeostasis and human health. Anything that affects or disrupts this equilibrium, such as molecules from foods, exposure to chemicals, or other types of stressors, has the potential to alter the functioning of the intestinal barrier and impact people’s health [18].

## 2. Inflammatory Bowel Disease

Several studies highlighted how GT alterations at various levels can compromise its functions and also lead to the onset of chronic diseases as well as inflammatory bowel disease (IBD) and autoimmune disorders [19,20].

Crohn’s disease (CD) is the most common subtype of IBD both in adults and in children, comprising 59–73% of pediatric IBD, which affects the entire GT [21]. Weight loss, watery diarrhea, and abdominal pain are the traditional symptoms of this illness. Particularly, the clinical signs and symptoms resemble those of allergic and infectious gastroenteritis. The pain is often of the colicky type and is more frequently found in the right lower quadrant of the stomach due to the increased involvement of the terminal ileum [22]. Most often watery, bloody diarrhea is more common when the colon and rectum are involved [23].

While CD is defined by dispersed “jumpy” lesions in the entire intestinal tract, from the mouth to the anus, typically in different evolutionary stages, ulcerative colitis (UC) is a recurrent and remitting IBD which is represented only by mucosal inflammation starting distally, in the rectum, and potentially spreading to cover the entire intestine (continuous ascending colitis) [24,25]. Yet, compared to CD, UC has a higher frequency of bleeding (83–95%). The mucous layer of the intestinal wall typically suffers superficial damage as a result of UC inflammation, and bloody diarrhea is the most common symptom in those who are affected [26].

Although there are still many unclear points about the pathophysiology of IBD, it appears to entail a number of factors, such as genetic predisposition, epigenetic or gene activity regulating factors [27], altered immune responses, environmental factors, and changed intestinal flora [26]. Recent studies underline the existence of an interesting connection between oxidative stress, gut microbiota, and immune response [28].

Reactive oxygen species (ROS), which is mainly responsible for oxidative stress, exert dual functions in a dose-dependent way: Low concentrations of ROS regulate cellular physiological processes, including redox signal transduction, gene expression, and receptor activation, which are beneficial for tissue turnover and cell proliferation [29], while higher concentrations of ROS can damage cellular molecules, such as DNA, proteins, and lipids, leading to cell senescence and even to death [30].

It is well known that the onset and progression of IBD are accompanied by persistent oxidative stress and inflammatory responses that have been correlated with dysbiosis of the gut microbiota [31]. External stimuli, including high-fat diet, smoking, circadian rhythms, and drug intervention, can induce oxidative stress which leads to gut dysbiosis [32]. Notably, IBD patients are characterized by a decreased gut microbial diversity and dysbiosis with a reduced proportion of Firmicutes and Bacteroidetes and an increased ratio of Proteobacteria [33]. GT of IBD patients contains lower levels of beneficial bacteria, such as Bacteroides, Lactobacillus, and Eubacterium, [34] and a concomitant increase in pathogenic species, including Enteroinvasive *Escherichia coli* (EIEC) [35]. The related dysbiosis has been correlated with the induction of oxidative stress and inflammation that are responsible for intestinal epithelium damage and increased intestinal permeability [36]. This, in turn, results in the release of harmful bacterial metabolic toxins into the blood, such as lipopolysaccharide (LPS) and pro-inflammatory chemokines and cytokines like tumor necrosis factor α (TNF-α), interleukin-1β (IL-1β), and interferon-γ (IFN-γ) [32], causing a worsening of the inflammatory process that characterizes IBD patients [36].

Oxidative stress induced by dysbiosis can act as upstream stimuli to evoke aberrant activation of the intestinal immune system, causing destruction of tight junction and damage to the intestinal mucosal barrier with antimicrobial peptide secretion. The release of pro-inflammatory factors (e.g., TNF-α, IL-6, and IL-1β) and the activation of pro-inflammatory enzymes (e.g., iNOS, COX-2, and NOX) [37] via nuclear factor-κB (NF-κB), JAK/STAT, and mitogen-activated protein kinases (MAPKs) signaling pathways [38] lead to a further increase in the level of ROS in the GT [39] and remodeling of the gut microbiota, in a vicious cycle in which dysbiosis and inflammation support each other. Restoring the homeostasis of the gut microbiota of IBD patients in favor of beneficial versus pathogenic bacterial species could be helpful in relieving oxidative stress and inflammation. Thus, ROS are considered among the main causes of IBD pathogenesis and a potential therapeutic target to limit its progression and alleviate its symptoms.

## 3. Potential Beneficial Effects of Dietary Polyphenols in IBD

Conventional IBD treatments are intended to turn off the inflammation in the acute phase of the disease and/or extend remission periods, and these include aminosalicylates, corticosteroids, antibiotics, immunosuppressive agents, and biologics [40], such as therapeutic TNF-α antagonists. These latter are able to restore the gut barrier [41], demonstrating a crucial role for TNF-α in the pathogenesis of IBD, due to its pro-inflammatory and pro-apoptotic effects.

These treatments could be effective for some patients, but many of them could lose the response over time or do not respond at all, not to mention their dangerous side effects, such as osteoporosis for corticosteroids [42], or increased risk of infection and gastrointestinal distress for immunosuppressants and aminosalicylates, respectively [40]. Finally, biologics, although very effective, are never first-line medications because of their high-cost [43], and often, IBD patients require lifelong treatment.

Collectively, novel therapeutic strategies for IBD treatment able to reduce oxidative stress and inflammation are urgently required. In this scenario, polyphenols are among the most suitable candidates.

Polyphenols are characterized by a structure composed of multiple phenolic units, and they are introduced with a diet made of vegetables, fruits, nuts, and beverages, such as coffee, tea, wine, and beer [44,45]. They could have very low molecular weight, such as coumarins, or they could be polymerized compounds, such as tannins, and conjugated or not with one or more sugar residues as glycosides. However, due to the presence of the hydroxyl group in their structure, polyphenols can scavenge ROS by hydrogen atom transfer or single-electron transfer reactions [46], having an important antioxidant and anti-inflammatory activity.

Several experimental evidences have linked polyphenol dietary intake to a reduced risk for cancer [47], obesity [48,49,50], type 2 diabetes [51], and metabolic syndrome [52]. In recent years, multiple in vitro and in vivo studies supported the hypothesis of a protective role for polyphenols administration in the IBD [53], by modulating the expression of transcription factor inflammatory targets, oxidative stress, intestinal barrier function, pathogen defense, and the composition of gut microbiota and its metabolite production, crucial for maintaining intestinal homeostasis.

The ultimate outcome of these processes appears to be a reduction in the pro-inflammatory cytokine storm that characterizes IBD. Larussa T. et al. analyzed biopsies obtained during colonoscopy from 14 patients with active UC, which were immediately placed in an organ culture chamber and treated with LPS from *Escherichia coli*, in the presence or absence of oleuropein (OLE), a glycosylated phenolic compound found in green olive skin, flesh, seeds, and leaves. The expression of IL-17 and COX-2 was significantly lower in the OLE-treated samples. Furthermore, OLE-treated colonic samples showed signs of reduced mucosal inflammation, with lower infiltration of B and T lymphocytes [54]. Similar effects were obtained in an ex vivo model of CD, in which exposure of intestinal biopsies to extracts from Olea europaea leaves reduced the production of pro-inflammatory mediators (IL-1β, IL-6, IL-8, and TNF-α) [55]. 

A recent metanalysis analyzed the effects of polyphenols from olives on murine models of UC. The authors proved that, although the use of these substances was not able to effectively prevent the development of the disease, in most UC models, olives constituents guaranteed a milder expression of the disease, better weight maintenance, and a reduction in rectal bleeding. Of note, according to the original aim of this study, the metanalysis should include both animal and human trials, but no human study met the eligibility criteria [56]. To date, only very few studies analyzed the effects of polyphenols intake in patients suffering from UC [57,58,59,60,61,62,63,64] and none on CD. Most of these considered the use of curcumin either as oral intake or as enema, others used sylmarine or resveratrol. The results obtained from these studies are controversial, regardless of the compound used; some of them demonstrate an improvement in both UC clinical [65] and inflammatory [62] parameters, while others show no effect [59]. These studies suffer from a great degree of heterogenicity, with different outcomes analyzed, different compounds used, variable concentrations and way of administration, different UC activity status, and, finally, a low number of enrolled patients. 

Other studies enquired about the effectiveness of polyphenols on IBD; however, these used products contain several polyphenols, as well as other substances like extra virgin olive oil [66] and mango pulp [67]; as a result, no specific conclusion can be drawn. Moreover, limiting the potential effectiveness of polyphenols’ anti-inflammatory effects, many polyphenols are poorly absorbed, resulting in their poor bioavailability. Thus, an increase in intestinal mucosal exposure, the target tissue of IBD treatment, is required. In addition, in the intestinal lumen, polyphenols interact with intestinal microbiota that modify them, altering their pharmacokinetics and producing metabolites which are completely different from the polyphenols that generated them [68] and with greater relevance and bioavailability than their precursors [31].

Finally, prebiotic-like properties of polyphenols increase the relative amount of intestinal beneficial bacteria, strengthening their anti-inflammatory activity [69,70] and antioxidant capability [71]. In summary, some studies proved the potential efficacy of polyphenols in reducing the inflammatory cascade (Table 1) (see the Section 4) typical of IBD, allowing us to hypothesize a role of polyphenols enriched diet to control symptoms and improve histology in patients. However, scientific evidence is far from allowing a clear indication of polyphenols intake as an “add-on therapy” for the treatment of IBD, with studies proving conflicting results. In addition, most of the data come from in vitro, ex vivo, and animal studies which are not easily comparable with the others, because of (1) the use of different models of colitis; (2) the use of different types of polyphenols, alone or in combination; (3) the use of concentration of polyphenols which might be difficult to obtain in human models; and (4) differences in measured outcomes. Similarly, human studies also showed several limitations: (1) studies conducted only in UC patients; (2) low number of subjects included; (3) high rates of drop-out; (4) lack of uniformity between the types of patients included (IBD in active and non-active phase, severity of colitis, etc.); (5) polyphenols method of administration (oral, enema, etc.); (6) use of different types of polyphenols, alone or in combination; (7) high variability in polyphenol’s dosages; (8) differences in measured outcomes; and (9) absence of an adequate study of the interactions between intestinal mucosa, gut microbiota, and concentration and bioavailability of polyphenols.

As already reported, we must emphasize that the putative effectiveness of polyphenols in IBD in in vitro, ex vivo, and animal studies is usually obtained with a concentration which cannot be achieved by a human’s normal diet. Thus, it is possible to hypothesize that polyphenols should be administered separately from normal nutrition, as concentrated extract (tablets, capsules, powders, etc.), either systemic or local. If it is known that physiological intake of polyphenols usually has no adverse effects on human health, no definitive conclusion has been drawn about the safety of high concentration of polyphenols intake. In fact, studies have shown that high concentration of polyphenols might (1) reduce intestinal iron absorption (chelating activity); (2) impair digestive enzymes (amylases, proteases, lipases, etc.) activity; (3) modify physiologic gut microbiota, allowing overgrowth of pathogen bacterial strains; (4) interact with drug-metabolizing enzymes, modifying their bioavailability; (5) act as prooxidants in specific environmental conditions, such as alkaline PH, high levels of metals and O_2_; and (6) stimulate mutagenesis by both prooxidant effect and interacting with DNA topoisomerases [72].

Considering all these limits, but also the undeniable potential of polyphenols in the treatment of IBD, it is absolutely necessary that research groups coordinate their efforts to remove the veil of uncertainty that still remains.

**Table 1 ijms-24-14619-t001:** Effects of some polyphenols on in vivo IBD models.

Polyphenol	Animal Model	Anti-Inflammatory Effects	Colitis-Inducer	References
Acacetin (5,7-dihydroxy-4′-methoxyflavone)	Mouse	It downregulates mRNA and protein expression of important pro-inflammatory cytokines (IL-1, IL-6, and TNF-α), as well as enzymes like COX-2 and inducible nitric oxide synthase (iNOS)	Dextran Sulphate Sodium (DSS)	Ren J et al., 2020 [73]
Oleuropein	Rat	It has both anti-inflammatory and antioxidant effects because it prevents the generation of pro-inflammatory cytokines, which lowers the levels of ROS in ulcers	Intrarectal Acetic Acid 4%	Motawea M. H. et al., 2020 [74]
Hydroxytyrosol	Rat	It significantly lowers colon malondialdehyde (MDA), myeloperoxidase (MPO), and nitric oxide (NO) levels while significantly increases superoxide dismutase (SOD), catalase (CAT), and glutathione peroxidase (GPX) levels. It also downregulates pro-inflammatory cytokines, reducing oxidative stress and inflammation in colon tissue	Intrarectal Acetic Acid 4%	Elmaksoud H. A. A. et al., 2021 [75]
Ellagic acid	Rat	It lowers colitis severity, by reducing colonic MPO activity. This effect is more potent when ellagic acid is contained in microspheres	DSS 3%	Ogawa Y. et al., 2002 [76]
Rutine (3-O-rhamnosyl-glucosyl-quercetin)	Rat	It lowers colitis severity and colonic MPO activity, with a dose-dependent effect	Rectally Trinitrobenzene Sulfonic Acid (TNBS) 15 mg	Kim H. et al., 2005 [77]
Epigallocatechin-3-gallate	Rat	It lowers colitis severity by reducing colonic MPO activity and enhancing SOD activity	TNBS 24 mg	Rat Mochizuki M. et al., 2010 [78]
Curcumin	Mouse	It lowers colitis severity by reducing colonic CD4^+^ T-cell infiltration and NF-κB activation and decreasing colonic IL-6, IL-12, IFN-γ, and TNF-α mRNA expression	TNBS 2.5 mg	Sugimotoet K. et al., 2002 [79]
Curcumin	Mouse	It lowers colitis severity by reducing colonic MPO activity and IL-1β level and suppressing NF-κB and p38 MAPK activation	Dinitrobenzene Sulfonic Acid (DNBS) 6 mg	Salh B. et al., 2003 [80]
Curcumin	Rat	It lowers colitis severity by suppressing NF-κB activation, blocking IκB degradation, and reducing IL-1β mRNA expression	TNBS 50 mg	Jian Y. T. et al., 2005 [81]
**Curcumin**	**Human**	**It lowers colitis severity by improving both clinical activity index (CAI) and endoscopic index (EI)**	**Non stimulated**	**Hanai H. et al., 2006** [65]

## 4. Molecular Mechanisms Underlying the Effects of Dietary Polyphenols in IBD

A great number of studies have shown that polyphenolic compounds showed inhibitory effects on NF-κB, an inducible transcription factor which regulates a large array of genes involved in different inflammatory processes, including flavones, isoflavones, flavonols, flavanones, chalcones, anthocyanins, lignans, stilbenes, and phenolic acids. Furthermore, many of them also positively regulate antioxidant signaling pathways, such as Nrf2 [82,83], to intensify the intestinal mucosal barrier, which is also beneficial for the homeostasis of the gut microbiota [84].

NF-κB, responsible for the transcription of several pro-inflammatory genes, including those for pro-inflammatory cytokines, (e.g., TNF-α and IFN-γ), has been found markedly overactivated in IBD, as evidenced by the increase in NF-κB p65 expression in biopsied tissue from IBD patients, particularly CD [85]. Thus, some authors have reported a direct and positive correlation between NF-κB activation in IBD patients and chronic mucosal inflammation [86]. The same pro-inflammatory cytokines, microbial endotoxins (e.g., LPS), and ROS [87] may be responsible for NF-κB activation, which is mediated by the phosphorylation of IkB, its inhibitory subunit, by a specific IkB kinase (IKK). Once phosphorylated, IkB is degraded and NF-κB is free to move to the nucleus of immune cells and enterocytes where it induces the transcription of its target genes [87].

The NF-κB inhibitory effects of polyphenols were described largely in vitro [88] by comparing 36 flavonoids on NF-κB activation in murine J774 macrophages. Daidzein, genistein, isorhamnetin, kaempferol, quercetin, naringenin, and pelargonidin inhibited NF-κB activation, reducing nitric oxide (NO) production, whereas stilbenes, such as resveratrol [89] and lignans, including arctigenin and demethyltraxillagenin [90], inhibited NF-κB by similar mechanisms but with different degrees of potency. All these findings have suggested the idea that polyphenols may be more potent anti-inflammatory agents than some conventional IBD therapies, such as 5-aminosalicylic acid, a common IBD anti-inflammatory drug, since they were effective already at lower concentrations (micromolar) than the others (millimolar) [91].

Most of the polyphenols act on the IKK complex, preventing IkB phosphorylation and NF-kB translocation to the nucleus. More specifically, some of them, such as apigenin and quercetin, mediate the inhibition of the IKK-γ-regulatory subunit [92,93], whereas other flavonoids, such as morin [94], fisetin [95], and gossypin [96], inhibit the IKK-β-regulatory subunit [97,98]. A further mechanism of control of NF-κB activity involves its phosphorylation on the p50 subunit, via phosphatidylinositol 3-kinase (PI3K)/Akt pathway, influencing NF-κB DNA binding and its transcriptional activity. Several flavonoids have been shown to inhibit this pathway through the inhibition of Akt or PI3K [97,99].

Toll-like receptors (TLRs) are receptors through which immune cells, but also epithelial, endothelial cells, and fibroblasts, recognize microbial molecules in order to protect us from pathogens or prevent an inflammatory response aberrantly triggered by commensal microbes [100].

Intestinal mucosa of IBD patients showed an elevated expression of LPS binding toll-like receptor 4 (TLR4), suggesting its involvement in the inflammation characterizing IBD patients. In fact, there is a close connection between TLRs and the production of pro-inflammatory cytokines [101]. Once activated, TLR signaling converges on NF-kB activation, which, in turn, leads to the transcription of pro-inflammatory genes [100], through two distinct signaling pathways, the MyD88-dependent and the TRIF-dependent pathways, that lead to the activation of the kinases TBK1 and RIPK1. Some polyphenols, such as Epigallocatechin-3-gallate (EGCG), act on both pathways suppressing the kinase activity of TBK1, a kinase whose activation leads to the production of Interferon type I inflammatory cytokines, as well as inhibiting IKK in murine RAW264.7 macrophages [102]. Other polyphenols, such as luteolin [103] and resveratrol [104], inhibit only the TRIF-dependent pathway through TBK1 inhibition.

Flavonoids have been reported to modulate MAPKs that influence NF-κB activity being involved in pro-inflammatory cytokines cellular production [105].

The mammalian MAPK family consists of three subfamilies: extracellular signal-regulated kinases (ERK), c-Jun N-terminal kinases (JNK), and p38 MAPKs (p38). Some flavonoids, such as quercetin, perform their anti-inflammatory functions by inhibiting ERK and JNK, while JNK and p38 are inhibited by catechin in stimulated THP-1 cells [106]. Fisetin attenuates phosphorylation of Akt and p38, but not of ERK and JNK, in dextran sulfate sodium (DSS)-induced colitis in mice [107], and proanthocyanidins act on JNK, ERK, and PI3K/Akt phosphorylation in a rat hepatic stellate cell line [108]. Finally, quercetin [109], kaempferol [110], luteolin [111], apigenin [112], and EGCG [113] have been reported to function by modulating only one or all the three kinases together. 

Nrf2 is the main transcriptional regulator of several detoxification and antioxidant enzymes belonging to the basic leucine zipper transcription factor family, which binds to the antioxidant response element (ARE) of DNA initiating transcription of many Phase II detoxifying and antioxidant genes. Given its function, Nrf2 regulates different processes, such as metabolism, inflammation, immunity, and autophagy [114]. It has two different localizations: In the cytosol, it is bound to its repressor Kelch-like ECH-associated protein (Keap1), which tags Nrf2 for ubiquitination maintaining it at low levels [115]. When cellular oxidative stress occurs, Keap 1 activity is decreased and Nrf2 is free to move to the nucleus where it binds to ARE inducing the transcription of its target genes for detoxifying and antioxidant enzymes, such as superoxide dismutase (SOD), glutathione S-transferase (GST), UDP-glucuronyl transferase (UGT), and NAD(P)H: quinone oxidoreductase (NQO1).

Polyphenols regulate the Nrf2-Keap1-ARE pathway both in a Keap1-dependent and independent manner [116]. Polyphenols-dependent protection from oxidative stress involves the induction of conformational changes in cysteine sulfhydryl residues of Keap1 that increase Nrf2 stability and accumulation, preventing its ubiquitination [117]. Resveratrol [118], quercetin [119], baicalein [120,121], xanthohumol [122,123], and others [116] have been shown to induce Nrf2 in vitro. Alternatively, polyphenols act in a Keap1-independent way, modifying protein kinases that phosphorylate Nrf2 at different sites, such as Ser40 and Ser408, resulting in stability increase and activation [124,125,126]. 

In addition, MAPKs can also influence Nrf2 activation. Some polyphenols, such as quercetin [127], resveratrol [128], lycopene [129], luteolin [130], procyanidins [131], anthocyanins [132], hesperidin [133], EGCG [134], and epicatechin [135], seem to be responsible for the upregulation of the Nrf2-ARE pathway through ERK; quercetin [127], lycopene [129], EGCG [136], and procyanidins [137] through p38 and kaempferol [138] and sappanchalcone [139] through JNK. Finally, resveratrol [128], EGCG [134], epicatechins [135], and procyanidins [137] activate the Nrf2-ARE pathway via phosphatidylinositol 3-kinases (PI3K).

NF-κB [116] can also modulate Nrf2-ARE activity. The two signaling pathways interact with each other [140], suggesting that the inhibition of NF-κB induced by polyphenols may also influence the activation of the Nrf2-ARE pathway [141] (Figure 1).

## 5. Foodomics Approach to Investigate the Relationship between Food and Health

Nowadays, food is increasingly considered not only an important energy source but also a crucial factor in maintaining good health and reducing the risk of disease. According to Hippocrates’s sentence “Let food be thy medicine and medicine be thy food”, today, the importance of healthy eating is widely recognized and it can also be demonstrated by highlighting the mechanisms underlying these health effects [142].

Food science, which comprises food chemistry and food microbiology, has made several steps forward in the development of new food products with longer expiration dates, improved processes of production and packaging, and better organoleptic characteristics, taking great advantage of the new analytical methods. These new analytical methods are those related to the holistic omics major types of technologies (genomics, transcriptomics, proteomics, and metabolomics), which gave life to a variety of different omics subdisciplines (epigenomics, lipidomics, interactomics, metallomics, diseasomics, etc.); each of them having its own set of tools, techniques, reagents, and software, making it easier for the researchers to connect food nutrients, introduced with the diet, to the health and disease of the individual [143].

These technologies differ not only in the laboratory techniques performed but also in the analysis of the biological activities of food components, being very helpful for developing novel biomarkers and for exploring how diet can affect gene transcription, protein expression, and the entire human metabolism [144].

Foodomics is a scientific discipline that aims to improve the health of the individual by studying the Food and Nutrition domains as a single domain using omics technologies, including nutrigenomics and nutrigenetics [145].

The genomics approach of Nutrition Science, which studies DNA structure and function, tries to clarify how our genome could modify the different individual responses to foods, whereas the genomics approach of Food Science aims to develop genetically modified harvests and livestock with a higher growth performance, infection/disease resistance, and so a consequent improvement of food nutritional values [146]. So, we can refer to a new type of comprehensive approach called the “foodomics approach” allowing us to look at the problem from the new perspective of using food science to improve human nutrition and its effects on human health. Our body can be influenced by the diet, as an external stimulus, in a positive or negative way, being a protective or a risk factor against certain types of human disease. Similarly, the diet can be influenced by external stimuli because we feed on meat, coming from living organisms, and vegetables which may be affected by breeding and agricultural techniques, respectively. Thus, the foodomics approach helps us to relate all these factors to each other with the aim of improving human health.

Genomics tools include the sequencing, assembling, and analysis of genome structure and function within an organism. Genomics technologies are intended to lead to knowledge of as many genetic sequences as possible through different technologies such as high-density arrays of oligonucleotides or complementary DNAs (cDNAs) and next-generation sequencing (NGS) technology. Each plant or farm animal gene composition can be well determined, leading to identifying specific marker traits and beneficial alleles responsible for healthier food properties [147]. In fact, genomics technologies allow us to put on a “graphic genotype” which plant growers can use to find inheritable chromosome sections and to identify some beneficial marker traits and gene alleles leading to the selection of crops with greater nutritional and safety characteristics. 

Genetics is the most common factor contributing to variability in the response to nutrition, as well as to eye and hair color [148], in cases such as PKU, lactose intolerance [149], and metabolic syndrome [150]. However, unlike eye and hair color, nowadays, it is increasingly understood that genetics is unable to fully explain the response to food. Understanding how our genes interact with nutrients may lead to the prescription of tailor-made diets for each individual: For this purpose, nutrigenomics was introduced [151].

The transcriptomics approach studies all the RNA transcripts, or transcriptome, inside a single cell or a group of cells through gene expression microarrays and large RNA sequencing (RNA-Seq) [151]. This leads to the knowledge of how global gene expression is modulated by food and the role of single nutrients in inflammation, oxidative stress, and cancer prevention [152], with the aim of implementing the strategies of microbial mitigation [153]. Gene expression microarray technology, introduced in the 1990s and now widely used [154], is used for producing new medicines and identifying the presence of hazardous substances or contaminants in food, such as pesticides, estrogen-like chemicals, and dioxins, thus improving its quality and safety [155].

Finally, it also allows us to evaluate how each food component affects the expression of host genes and to study metabolic transformations at molecular levels [156].

Furthermore, proteomics tools involve the analysis of proteins in food, not only structurally or functionally but also elucidating their interactions and how their structure alterations could modify their role inside our cells [157]. Essentially, proteomics techniques are divided into extraction of proteins from cells, enzymatic digestion, molecular separation by two-dimensional electrophoresis (2-DE) or multi-dimensional liquid chromatography (MDLC), and mass spectrometric (MS) analysis [158]. 

Proteomics technologies, which make use of high-performance separation techniques together with high-resolution MS, are widely exploited in Food Science for monitoring food quality and the presence of microbial contaminants [159] and animal health, allowing identification and characterization of protein tissues or biological fluids [160].

Current metabolomic technologies are essential in both food and nutrition science [152]. Targeted metabolomics enables us to make precise chemical analyses of up to a thousand metabolites, investigate how they are affected by gene mutations causing protein changes, and characterize metabolic processes that underlie several pathologies with the goal of finding new functional food biomarkers or therapeutic targets for diseases, playing a main role in our metabolism [153].

The metabolomics approach employs techniques such as sample preparation, metabolite extraction, sample testing, selection and use of appropriate analytical tools, and collection of the data [154]. Metabolomics technologies include liquid and gas chromatography-MS (LC-MS and GC-MS), nuclear magnetic resonance (NMR), and capillary electrophoresis-MS (CE-MS) [155]. Despite having a very low sensitivity, NMR is mostly used for functional food studies aimed at the characterization of active ingredients as well as the effects of different biomarkers [156] through the quantization of metabolites and the analysis of their structure in detail. Requiring small sample sizes and simple preparation procedures [157], MS technologies are especially used to characterize and quantify unknown metabolites, being very fast in separating them with high sensitivity and determining the composition and origin of foods during various processes of manufacturing [158]. In addition, metabolomics technologies are useful for detecting individual metabolic changes and degraded food [160] and, in epidemiological studies in the field of Nutrition Science, for characterizing novel biomarkers (Table 2).

## 6. Trimethylamine-N-Oxide: A Novel Biomarker of Inflammation and IBD Diagnosis

For a long time, trimethylamine-N-oxide (TMAO) was considered only a waste product of the metabolism of choline without any function but, surprisingly, nowadays there is increasingly convincing evidence suggesting a positive correlation between TMAO plasma levels and inflammatory diseases, such as atherosclerosis and consequent cardiovascular diseases (CVDs), and tumors [161]. TMAO is generated by a two-step process: first, the choline present in foods, such as beef liver, eggs, beef, chicken, other types of meat and poultry, milk, beans, edamame, legumes and soy products, mushrooms, fish, cauliflower, broccoli, asparagus, quinoa, and avocados, is converted to TMA by choline TMA lyase, an enzyme characterizing bacteria from gut microenvironment. Once produced, TMA enters the portal circulation where it is rapidly oxidized in the liver by hepatic flavin monooxygenases (FMO), forming TMAO, which is then mainly excreted in the urine [162,163]. 

FMO3 is the most represented FMO enzyme in humans [164,165]. Genetic deficiency in FMO3 is reported to predispose to fish odor syndrome, a rare condition resulting from the failure to convert TMA to TMAO [166]. Humans ingest a small amount of TMA with food whereas most are formed from dietary phosphatidylcholine and carnitine that are metabolized by anaerobes or facultative anaerobes bacteria species colonizing our GT [167], such as Clostridia, Proteus, Shigella, and Aerobacter [162].

Supporting this, Tang et al. recently demonstrated that GT microbiota is responsible for producing TMAO from phosphatidylcholine present in food [168]. After subjecting individuals to a high-choline meal, referred to as a “choline challenge”, TMAO plasma levels were measured over an 8-h period. An increase in TMAO plasma levels was observed within the first hour of meal intake. When antibiotics for gut microbiota suppression were given to the same subjects before a second “choline challenge” was administered, TMAO plasma levels disappeared, leading them to hypothesize that TMAO plasma concentration is dependent on gut microbiota and alterations of its composition, in terms of the presence of bacteria species, could lead to its variation. Some metabolites produced by gut microbiota during the degradation of nutrients present in the diet [169] can trigger an inflammatory response, thus activating the immune system improperly, often causing the onset of inflammation and related diseases [170,171].

Recently, evidence obtained from vascular inflammation models strongly supports the hypothesis that TMAO can induce inflammation and immune regulation by directly inducing the expression of TNFα, NLRP3 inflammasome, mitochondrial ROS, and NF-κB, which are critical pro-inflammatory mediators, and down-regulating anti-inflammatory cytokines, such as IL-10 [172,173]. By contrast, high plasma TMAO correlates with improved efficacy of immunotherapy in several types of tumors, boosting antitumor immune responses, thereby rendering pancreatic ductal adenocarcinoma (PDAC) [174] more responsive to checkpoint blockade immunotherapy (ICB) (anti-PD1 and/or anti-Tim3) as well as for triple-negative breast cancer (TNBC), probably in the first case potentiating the type-I interferon (IFN) pathway, while in the second activating the endoplasmic reticulum (ER) stress kinase PERK, which induced gasdermin E-mediated pyroptosis in tumor cells and enhanced CD8+ T-cell-mediated antitumor immunity in TNBC in vivo. Thus, since the production of TMAO by the gut microbiome could be regulated through an increased intake of choline with food or using TMA lyase inhibitors, it turned out to be an excellent therapeutic target to be induced for the enhancement of the effectiveness of cancer treatment [175,176] and to be inhibited for the treatment of inflammatory diseases, being used also as a diagnostic tool, as for IBD.

It is well known that in IBD patients there is a loss of microbial diversity in the inflamed gut [177,178]; however, it is still unclear how this loss impacts IBD activity, phenotype, and severity. It is noteworthy that many studies have focused on the alteration of microbiota in disease, but very few are intended to use such alterations as biomarkers of diagnosis and progression of disease [179]. Today, IBD diagnosis involves screening for hematological features, such as inflammatory markers and elevated white blood cell count, but is confirmed only through endoscopic examination with multiple biopsies [180].

Based on their findings, Wilson et al. reported that plasma TMAO levels were significantly decreased in IBD patients when compared to non-IBD controls, suggesting that its plasma levels measurement might be clinically relevant not only for IBD treatment but also for its noninvasive diagnosis [181].

## 7. Gastrointestinal Protective Effects Mediated by Polyphenols: Foodomics-Based Approaches

Today, omics technologies are a valuable resource for identifying polyphenols in foods and for analyzing the changes that they induce inside our cells at the molecular level when present in the diet. Thanks to foodomics [180] advanced technologies, nowadays, it is easier to identify bioactive compounds, like plant-derived polyphenols, and analyze them in fruit, drinks, grain, oil, and so on [182]. Specifically, gene-based genomics and transcriptomics can be used to study the interaction between polyphenols and the GT. Gene expression microarray technology is useful for investigating how genes interact with each other after the introduction of bioactive compounds in food [183]. Experimental evidence from gene expression microarray technology as reported by Valdès et al. showed that polyphenols from rosemary extract did not induce apoptosis in two colon adenocarcinoma cell lines [184]. Transcriptomics was used to evaluate the effects, at the molecular level, of red wine polyphenols on the colonic mucosa of F344 rats [185]. 

The authors, following the analysis of the expression of 5707 genes, demonstrated that red wine polyphenols improve the function of the colonic mucosa and possess an anticancer activity which is expressed through a reduction in oxidative stress, modulation in microbiota composition, and down-regulation of genes involved in metabolism, transport, and signaling transduction [185]. Transcriptomics studies were also performed by Wang et al. [186] using 16S ribosomal RNA (rRNA) amplicon sequencing (16S-Seq) and Shotgun metagenomic community sequencing (SMC-Seq) to investigate the interaction between green tea polyphenols and the gut microbiota of female Sprague–Dawley (SD) rats treated with green tea polyphenols for 6 months. Bacteroides and energy-metabolism-related genes in SD rats treated with green tea polyphenols were modified in a dose-dependent way, suggesting beneficial effects for consumers. 

Data reported by Yang et al. [187] have shown how a combined action of different polyphenols influences gut microbiota diversity and structure during mice colitis-related carcinogenesis (CRC) and how these changes, in turn, influence the success of treatment having a regulatory effect on 17 signaling pathways involved in related genes, including several biomarkers associated with CRC, such as COX-2, EMR1, PCNA, and caspase-3, which were strongly changed by polyphenol treatment. Also, proteomics was used to clarify the interaction between probiotics and GT and the functional protein changes of probiotics that they induced, as well as to identify stress adaptation markers to better understand some probiotics’ features in food and also their ability to overcome the GT considering their immunomodulatory activity, colonization, and host interaction [188]. 

A study on the effect of polyphenol-rich rosemary extract on HT-29 human colon cancer cells was reported by Valdes et al. [189], based on nanoliquid chromatography–tandem mass spectrometry (nano-LCMS/MS) in combination with stable isotope dimethyl labeling (DML) technology. The authors found an antiproliferative effect exerted by this extract characterized by an increase in autophagy. Moreover, several altered proteins found are involved in the activation of Nrf2 and the unfolded protein response (UPR). Other studies used microarray technology and proteomics approach to evaluate the reduction in intestinal inflammation in a mouse model of IBD induced by green tea extract rich in polyphenols. This extract was able to reduce transcripts and proteins related to immune and inflammatory response pathways as well as increase those related to xenobiotic metabolism pathways. These effects seem to be regulated by peroxisome-proliferator-activated receptor-α (PPAR-α) and signal transducer and activator of transcription 1 (STAT1) [190]. On the other hand, the antibacterial activity of catechin on *Escherichia coli* O157:H7 cell lines in vitro resulted in various changes in the protein expression related to cell structure and processing of genetic information [191].

Metabolomics is an important part of foodomics, as it allows us to characterize biological functions and phenotypes [192], to target specific metabolic pathways and understand their molecular mechanisms. Therefore, how polyphenols interact with the GT was investigated through metabolomics approaches in terms of analysis of small molecule metabolites. Specifically, the phenolic extract from extra-virgin olive oil (EVOO-PE) was tested for its antiproliferative effects on SW480 and HT29 human colon cancer cell lines by nano LC-ESITOF-MS technology. Several metabolites of phenolic compounds, especially quercetin and oleuropein aglycones (and their derivatives), were present in the cytoplasm of SW480 and HT29 cells, where they induced apoptotic processes [193]. 

The protective effect of polyphenol-rich bee pollen (BP) extracts was also evaluated using UPLC-QTOF/MS metabolomics in the in vitro model of intestinal barrier Caco-2 cells during the early stages of dextran sulfate sodium (DSS)-induced colitis. Metabolites present in cells treated with BP, probably acting on the glycerophospholipid metabolic pathway, were totally different from those not treated, suggesting that BP suppresses inflammation-modulating cell metabolism. In particular, BP was able to up-regulate the mRNA expression levels of antioxidant factors, such as NQO1, Txnrd1, and Nrf2. In contrast, the mRNA expression of inflammatory factors including TNF-α and IL-6 was down-regulated, corresponding to the inhibition of MAPKs signaling [194]. Furthermore, transcriptomics and proteomics technologies together were used to study the anticarcinogenic effects of quercetin introduced with the diet on the colon mucosa F344 rats, an experimental model of colorectal cancer. Quercetin significantly down-regulated the oncogenic MAPK pathway, up-regulating the expression of cell cycle inhibitors, like MUTYH, and tumor-suppressor genes, such as PTEN, Tp53, and MSh2. Furthermore, dietary quercetin enhanced PPAR-α targets genes and increased the expression of genes involved in mitochondrial fatty acids (FAs) degradation [195].

In their studies, Di Nunzio et al. showed the anti-inflammatory potential of olive aqueous extract and its effects on the cell metabolome in the Caco-2 cell line. In both basal and inflamed conditions, the olive extract supplementation decreased not only the pro-inflammatory cytokine IL-8 secretion but also induced modification in the metabolome, leading to a change towards a glucose-saving program, with a subsequent decrease in appetite following the maintenance of anorexigenic hormone secretion [196]. A multi-omics approach proved to be the most suitable for understanding the possible effects of dietary polyphenols in GT disorders, which often involve both molecular and biochemical mechanisms in their metabolization. The omics together can be useful for characterizing genes, proteins, and metabolites involved in polyphenol metabolism by the intestinal microbiota and the anti-inflammatory or antiproliferative effects that their metabolites have in IBD or GT cancer, respectively. In addition, multiple omics techniques could help to better understand the integrative effects of diet, host, and microbiota in order to develop new therapeutic strategies to use in a personalized nutrition [197]. 

In this regard, transcriptomics, proteomics, and metabolomics technologies (microarray analysis, MALDI-TOF/TOF-MS and CE/LC-MS) were used together to evaluate the effectiveness of the antiproliferative effects of polyphenols extracted from rosemary on human HT29 colon cancer cells. Based on the combination and comparison of different analytical platforms and in response to molecular changes induced by food ingredients, these studies demonstrated the antitumor effects of polyphenols, increasing interest in using integrative strategies, such as foodomics, to reduce cancer risk [198]. Finally, a study by Mayta-Apaza et al. demonstrated that polyphenol-rich tart cherries can increase the number of beneficial bacteria in the gut through their metabolites using bacterial fermentation assays performed on concentrate juices or pure polyphenols. Combining 16S rRNA gene sequence and metabolomics experiments showed that polyphenols, in vitro, were metabolized by the gut microbiota into 4-hydroxyphenylpropionic acids, leading to an increase in Bacteroides. However, in in vivo data, there was a decrease in both Bacteroides and Bifidobacterium [199], suggesting that in vitro and in vivo data obtained using a foodomics approach to evaluate the effects of polyphenolic extracts from natural foods or single polyphenolic compounds do not always match. This is because polyphenols’ digestive process and their metabolization by gut microbiota are very complex and in vitro experiments performed using integrating foodomics on colon cell lines or mice models can lead to a reduction in the in vivo physiological variability that characterizes their behavior. Polyphenols present in different natural foods vary significantly, and further research on their biological activities and studies at multiple molecular levels of genes, proteins, and metabolites are required to fully understand their effect on GT and to assess their effectiveness in the treatment of IBD.

## 8. Low Bioavailability and Bioaccessibility, Appropriate Doses, and Side Effects of Polyphenols

Adequate bioavailability and bioaccessibility are key problems limiting the usage and effectiveness of polyphenols for the treatment of different diseases. The bioaccessibility of polyphenols is associated with the amount of compound accessible for absorption [200] that is different from that introduced by oral administration, depending on the site of the pathological process. In the colon, the distal part of the digestive tract is affected during UC and low amounts of polyphenols are delivered, resulting in low bioaccessibility. Increased bioaccessibility is found in diseases affecting the upper parts of the digestive tract, such as in CD, mainly the small intestine. Polyphenols exhibit low bioavailability, decreasing from phenolic acids to isoflavones, flavonols, catechins, flavanones, proanthocyanidins, and anthocyanins [201], probably caused by the same factors that reduce their bioaccessibility. An intrarectally administration can be carried out to overcome both problems affecting oral administration, which still remains the preferred and most widely used route of administration for its non-invasiveness, low cost, and high patient compliance. 

Furthermore, polyphenols can also have adverse effects, as reported by some in vivo studies, including nephrotoxicity, hepatotoxicity, and iron deficiency, especially at high doses [202]. As recently reported by Lambert et al., high doses of EGCG can induce hepatotoxicity by generating increased amounts of malonyldialdehyde (MDA) and 4-hydroxynonenal (4-HNE) [203,204]. Moreover, EGCG was also found to enhance the expression levels of pro-matrix metalloproteinase-7 by inducing oxidative stress in HT-29 human colorectal cancer cells [205].

Tea is the most consumed beverage in the world, beyond water. In the last few years, several consumers and researchers have focused their attention on green tea for its potential beneficial effect on health. Kim et al. reported that a diet enriched with green tea polyphenols (GTP) at 1% enhances pro-inflammatory cytokines, consequently aggravating colitis in DSS-exposed colons and promoting colon carcinogenesis, and causes instead a reduction in superoxide dismutase (SOD) and catalase activities in non-treated mice [206]. Several human cases of hepatotoxicity following consumption of dietary supplementation containing green tea extracts have been reported by Mazzanti et al. [207]. In only 7 of the 34 reported cases, hepatotoxicity has been related to concomitant medications, such as administration of diclofenac [208], paracetamol [209,210], or progestogens [211,212,213], which have been associated with some cases of toxic hepatitis [214]. The remaining 27 cases (80%) confirmed green tea as the main cause of hepatic damage, showing a temporal correlation between the consumption of green tea preparations and the appearance of the side effects.

Moreover, neuroprotective polyphenols used against amyloid β aggregation, their own pro-oxidant activity, are able to damage DNA and other macromolecules of normal cells, including endothelial cells, leading them to apoptosis and the onset of CVD [215]. By contrast, low concentrations of polyphenols can prevent hepatic and renal damage [216].

Further studies are therefore necessary to highlight the side effects occurring after administration of polyphenols and determine the most appropriate doses with a pro-health activity and without toxicity. In this regard, to increase their bioavailability and reduce the dose at which they are effective and so their possible side effects, novel bioformulations using polyphenols are being developed for IBD treatment, and encapsulation of polyphenolic compounds is strongly recommended, such as for resveratrol-encapsulated microsponges delivered by pectin-based matrix tablets that resulted more therapeutically effective than pure resveratrol in rats with colitis [217], as well as resveratrol delivered by silk fibroin nanoparticles in an experimental model of IBD in rats [218] and β-lactoglobulin nanospheres encapsulating resveratrol that alleviated inflammation in Winnie mice with spontaneous UC [219].

Furthermore, rosmarinic-acid-derived nanoparticles conjugated with poly(ethylene glycol), rosmarinic-acid-loaded nanovesicles, oleuropein-loaded lipid nanocarriers, silica-installed redox nanoparticles with silymarin (compound being flavonolignan), or nanoparticles with curcumin were reported to be effective in lowering colonic inflammation in IBD in vivo [220,221,222], not only modulating the expression of the genes involved in the inflammatory response in a rat model of UC but also improving the mucosal lesions and preserving the distribution of telocytes, interstitial cells with a crucial role in colonic tissue homeostasis [223]. Taken together, these findings suggest that the strong limitations for using polyphenols in the treatment of IBD can be overcome by creating encapsulates; however, little is still known, and research into synthesizing polyphenol encapsulates and assessing their utility in IBD is continually developing.

## 9. Precision Nutrition and IBD Prevention and Treatment

The exact knowledge and understanding of how nutrient metabolism can affect the response of our body to the diet have been highlighted by many researchers [224]. In this respect, metabolomic helps us to evaluate the impact of specific nutrients introduced with the diet on an individual’s health, enabling us to characterize new food-derived biomarkers and to understand how the same foods are metabolized differently by different individuals in healthy or unhealthy conditions, such as intolerances or allergies to some food components.

Precision nutrition (PN) is a relatively new discipline, and often, the term precision nutrition is used instead of personalized nutrition [225] as these terms are often used interchangeably; however, a distinction between the two is increasingly needed [226]. 

Biological variability between individuals in response to nutrition is the basis of PN [227]. Hence, knowing the factors causing this variability and its effect, we could predict the outcome and translate it into nutrition advice. This area of nutrition focuses on the effects of the nutrients over the genome, proteome, and metabolome [228], trying to clarify how gene expression may be affected by nutrients introduced with the diet, such as polyphenols and other bioactive compounds. Thus, the PN is closely related to a deep “metabolic phenotyping” achieved through omics technologies, including polymorphisms and genetic variants analysis (genomics), DNA methylation and histone modifications analysis (epigenomics), evaluation of microbiota composition (epigenomics), the complete set of RNA transcripts and protein analysis (transcriptomics and proteomics), and the study of chemical processes involving metabolites, intermediates, and products of cell metabolism (metabolomics) (Figure 2). 

One of the earliest metabolomics methods centered on the investigation of single nucleotide polymorphisms that affect illnesses specifically linked to the metabolic state. Multiple polymorphisms can be now investigated in a single experiment, thanks to modern sequencing technology. Ongoing studies are evaluating the variations in responses to the same food patterns and concentrating on gene–diet interactions. This implies that many individuals respond differently to the same dietary ingredients (31–34). Analysis of the miRNA-410 gain-of-function mutant polymorphism (rs13702) in the lipoprotein lipase (LPL) 3′-untranslated region provides a stunning illustration. A Mediterranean-style diet decreased triglyceride concentrations and the risk of stroke, whereas this did not happen with the control diet, demonstrating a gene–diet interaction (31, 35). Another example concerns the intestinal microbiota, in particular, how the Mediterranean diet influences it, modifying both the species present and at the metagenomic level (36). The Mediterranean diet has several systemic advantages, particularly since it contains a lot of antioxidants like vitamin E, β-carotene, vitamin C, and flavonoids, as well as minerals like selenium and natural folate. Dietary antioxidants have been demonstrated to have positive benefits on coronary heart disease in the case-control INTERHEART trial. In fact, a lack of these could change lipoprotein oxidation, which would favor the development of atherosclerotic events. A recent clinical trial also found that extra-virgin olive oil supplementation together with a Mediterranean diet reduced levels of circulating oxidized low-density lipoprotein (LDL) and other inflammation-related indicators [229,230]. Therefore, nowadays, it is increasingly assigned a crucial role to the gene–diet interaction both in maintaining a state of physical well-being and in developing and preventing a pathology. The antioxidants, undiscussed protagonists of the Mediterranean diet, protect us from several chronic, and especially inflammatory, diseases by modulating gene expression [231].

Data related to genome variations and expressed through single nucleotide polymorphisms (SNPs) and other inherited genetic variations are collected and analyzed by nutritional specialists and related, in general, to food, lifestyle, and environment [232]. Therefore, IBD patients need PN plans which take into account individual biological, clinical, and lifestyle characteristics that are responsible for different food effects on disease outcomes, providing “therapeutic diets” that aim to improve their quality of life.

## 10. Conclusions

Nowadays, there is growing interest in evaluating individual responses to nutrients, a precision nutrition research initiative, and its association with improving IBD management and clinical endpoints. The long-term goal is the creation of predictive models, integrated with quantitative and qualitative measurement of the patient response, for customized IBD management, with the awareness that diet could be as effective as medical therapy, such as steroids treatment, especially for controlling and limiting inflammation that often characterize IBD patients. 

Notably, an elimination diet, already used as a first-line therapy for pediatric CD in many countries, is effective, safe, and inexpensive although it compromises the quality of life. Hence, it might be useful to identify, and eventually eliminate, only those foods that trigger unwanted inflammatory responses in the patient, maximizing the intake of specific nutrients deficient in many patients with CD and personalizing it according to the patient’s needs. 

It is well known that polyphenols, an important component of fruits and vegetables with pro-health properties, are able to ameliorate colon damage, restore gut microbiota dysbiosis, and even inhibit the colonic production of pro-inflammatory mediators characterizing IBD patients. In addition, they are able to enhance the expression levels of anti-inflammatory cytokines and antioxidant enzymes or reduce apoptosis of gut epithelial cells. Despite this, to date, very few clinical studies have assessed polyphenols effectiveness in clinical trials, which may be due to the limitations associated with their low bioavailability and bioaccessibility that can be overcome through the encapsulation of polyphenols, using micro- and nanocapsules, micro- and nanoemulsions, micro- and nanoparticles, or even more complex delivery systems.

Thus, different expertise is needed to design new therapeutic strategies based on the intake of polyphenols as a supplement to a personalized diet.

In the foodomics field, researchers from various disciplines including food chemistry, analytical chemistry, biochemistry, microbiology, molecular biology, food technology, clinical sciences, and other areas have to collaborate, with a multi-omics approach, in order to optimize human health and well-being.

Clinical trials with polyphenols for IBD are promising, even if their experimental design is complex due to the several mechanisms involved and the inherent difficulty in observing interactions between host metabolism, the gut microbiome, and the exposome in vivo. Metabolomics provides a comprehensive “snapshot” of small-molecule metabolites and together with microbiome sequencing could help us to elucidate functional changes within our organism. Together, multiomic integration methods may clarify the relationship between treatment and host, treatment and microbe, and the interactions between host and microbe, which play a key role in IBD pathogenesis and response to treatment. These relatively new foodomics approaches may provide critical insights to determine the clinical value of polyphenols for IBD therapy.

## Figures and Tables

**Figure 1 ijms-24-14619-f001:**
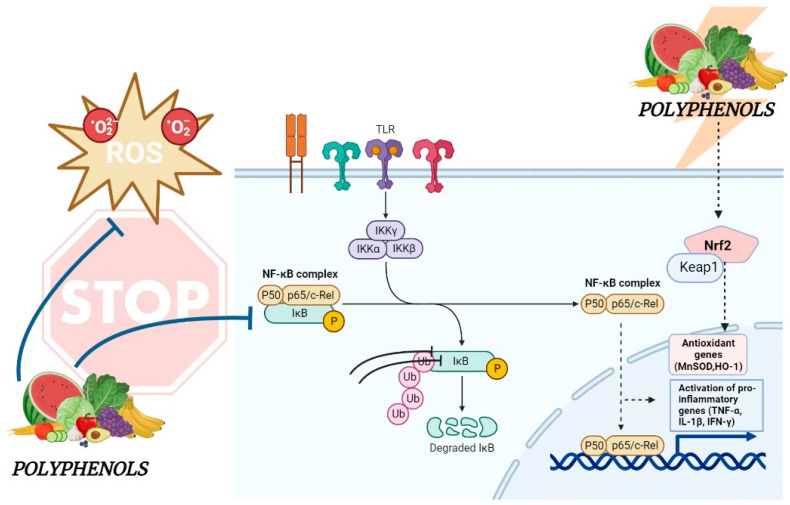
Molecular mechanisms modulated by dietary polyphenols in IBD. The figure summarizes the inhibitory effects of plant polyphenols on NF-kB (non-dashed line), a key transcription factor for pro-inflammatory genes, and the activating effects on Nrf2 (dashed line), the main transcriptional regulator of several detoxification and antioxidant enzymes. Modulation of these pathways from polyphenols could represent an alternative therapeutic strategy for IBD.

**Figure 2 ijms-24-14619-f002:**
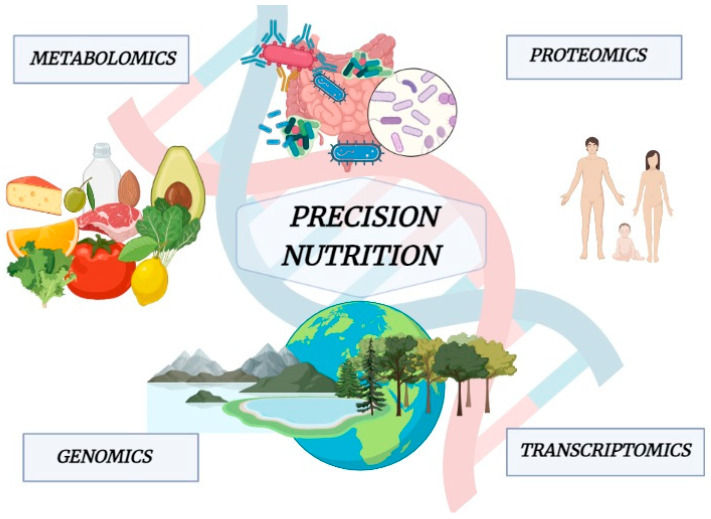
Multiomics explain the close relationship between host, foods, microbiota, microenvironment, and the biological variability between individuals in response to nutrition.

**Table 2 ijms-24-14619-t002:** Main omics types and subtypes technologies.

Omics Technologies		
Type	Tools	Focus
**Genomics**	- Next Generation Sequencing (NGS) - Genome-Wide Association Studies (GWAS) - Structural and Functional Analysis - Genomic Characterization - Bioinformatics	Genome complete sequences study by DNA sequencing, genetic profiling, recombinant DNA assays, structural and functional genome analysis
**Transcriptomics**	- RNA Sequencing - Microarray - qRT-PCR - Expressed Sequence Tag (EST)-Based Methods - Serial Analysis of Gene Expression (SAGE)	Total mRNA and gene expression study by RNA sequencing, expression profiling and transcriptional regulation analysis
**Proteomics**	- Mass Spectrometry (MS) - Protein Microarrays - Flow Cytometry - 2D-PAGE - Cytometry by Time of Flight (CyTOF)	Structural and functional proteins study by protein identification, quantification and post-translational modification, and expression profiling analysis
**Metabolomics**	- Nuclear Magnetic Resonance (NMR) - Gas Chromatography-Mass Spectrometry (GC-MS) - Liquid Chromatography-Mass Spectrometry (LC-MS)	Cellular metabolites study by metabolite and intermediates profiling, hormones and signaling molecules analysis
**SUBTYPE**	**FOCUS**	
**Epigenomics**	Epigenetic changes study in the regulation of genic expression and function by DNA methylation and histones modification analysis	
**Lipidomics**	Molecular characterization of cellular lipid and their biological roles by study of their pathways and networks in biological systems	
**Interactomics**	Study of the complex network of molecular interactions between proteins and other biological macromolecules which take place inside the cell	
**Metallomics**	Identification, distribution and interactions of metals and metalloids binding biomolecules, and their role in biological systems	
**Immunomics**	**Analysis of immune system regulation and response to infections by genome-wide techniques to identify antigens or epitopes linked to host immune response**	

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
