# Peer review of "Foodomics-Based Approaches Shed Light on the Potential Protective Effects of Polyphenols in Inflammatory Bowel Disease"

_ijms, 2023, doi:10.3390/ijms241914619_

Round 1

Reviewer 1 Report

To: International Journal of Molecular Sciences 

Dear EIC Prof. Dr. Maurizio Battino.

And dear section editor.

This is my review of manuscript ID: ijms-2597656.

The manuscript reviewed the mechanisms underlying IBD and some approaches to further understanding it. The title was previously commissioned. Especially the studies by Zhang et al. and He et al. However, I prepared some minor and major comments for this manuscript.

  • The authors should clearly announce the difference between their studies and previously published papers. This is very crucial for readers.
  • The conclusion is unclear. The authors are not allowed to use references in this section.
  • There are several misspellings of immunologic terms. For example, PRRs are called pattern recognition receptors for all microbial agents. In line 47, the authors used PRR for bacteria only. Kindly check the application of terms in the manuscript.
  • In Section 9, some of the contents were colored red. Check the writing.
  • The title of Section 5 is clear. But the titles of sections 6 to 9 are incomplete. For example, what is the relationship between the genomic approach and the content of its related section?

Author Response

  • The authors should clearly announce the difference between their studies and previously published papers. This is very crucial for readers.

We thank the reviewer for the comment. The effect of polyphenols in IBD has been previously reported in other studies. However, our manuscript is a review that speculates about the possible benefits of the use of polyphenols for IBD therapy also highlighting the potential risks and limitations of their use. The review also analyzes the importance of new foodomics approaches that, by assessing the biological effects of polyphenols in inflammatory bowel disease, can provide critical insight for a possible use of polyphenols as supplements for IBD therapy. These aspects are reported in the abstract and in the conclusion.

  • The conclusion is unclear. The authors are not allowed to use references in this section.

We modified the conclusion to make it clearer and we also deleted the references.

  • There are several misspellings of immunologic terms. For example, PRRs are called pattern recognition receptors for all microbial agents. In line 47, the authors used PRR for bacteria only. Kindly check the application of terms in the manuscript.

We thank the reviewer for the comment. We modified the terms in all the text as required.

  • In Section 9, some of the contents were colored red. Check the writing.

We deleted the red colored contents in Section 9 as suggested.

  • The title of Section 5 is clear. But the titles of sections 6 to 9 are incomplete. For example, what is the relationship between the genomic approach and the content of its related section?

We thank the reviewer for this comment. We modified the manuscript by joining the sections 6-8 in a single paragraph that discuss in a more concise manner the different omics approaches (genomic, transcriptomic, proteomic and metabolomic) that can be used to investigate the relationship between food and health.

Reviewer 2 Report

The present manuscript explains the omics tools used to assess the biological effects of polyphenols in inflammatory bowel disease. The uses of various omics tools and their applications were explained well. The mechanisms underlying the disease conditions and effects were well explained. The flow of manuscript was found to be good. The work was concluded well.

The flow and language was found to be good. No grammatical or spelling errors were found.

Author Response

The present manuscript explains the omics tools used to assess the biological effects of polyphenols in inflammatory bowel disease. The uses of various omics tools and their applications were explained well. The mechanisms underlying the disease conditions and effects were well explained. The flow of manuscript was found to be good. The work was concluded well.

We thank the Reviewer for appreciating the paper.

Reviewer 3 Report

This review explores the potential protective effect of polyphenols, natural antioxidants, in inflammatory bowel disease. It delves into the mechanisms underlying the health benefits of a nutritious diet and the development of new food products using foodomics-based approaches, including food chemistry and food microbiology. The article also introduces the use of holistic omics technologies, such as genomics, transcriptomics, proteomics, and metabolomics, to establish connections between food nutrients and individual health and disease. In conclusion, the article discusses the potential clinical implications of this research for patients with inflammatory bowel disease.

The structure of this review is complete and has good reference value. It is suggested that it be accepted and published after modification according to the following suggestions:

1.     The review could benefit from a more comprehensive discussion of the potential risks and side effects associated with high polyphenol consumption. It is essential to note that these risks can vary depending on the type and dosage of polyphenols.

2.     I encourage the authors to summarize and discuss the potential limitations of studies on polyphenols for treating IBD. This should encompass aspects such as small sample sizes and the lack of control for other dietary factors, which may affect the validity of the findings.

3.     It would be valuable for the authors to provide clear recommendations in the summary or a related section on how to incorporate polyphenols into the diets of IBD patients.

There are some minor grammatical errors that need to be corrected.

Author Response

The structure of this review is complete and has good reference value.

We thank the reviewer for appreciating the manuscript.

  1. The review could benefit from a more comprehensive discussion of the potential risks and side effects associated with high polyphenol consumption. It is essential to note that these risks can vary depending on the type and dosage of polyphenols.
  2. I encourage the authors to summarize and discuss the potential limitations of studies on polyphenols for treating IBD. This should encompass aspects such as small sample sizes and the lack of control for other dietary factors, which may affect the validity of the findings.
  3. It would be valuable for the authors to provide clear recommendations in the summary or a related section on how to incorporate polyphenols into the diets of IBD patients.

We thank the reviewer for his suggestions, which we welcome with great pleasure because they have allowed us to discuss the potential risks and side effects as well as the potential limitations of polyphenol consumption in IBD patients. In particular, we have added what was requested at the end of paragraph 3 (pag. 5 lines 212-246) “Potential beneficial effects of dietary polyphenols in IBD”, underlining not only the limits of in vitro, ex vivo and animal model studies, but also the limits of human ones (comment 2). And it is precisely from the examination of one of these limits 'Difficulty in reaching the polyphenols concentrations which proved effective in vitro through normal human nutrition', that we responded to both the first and third comments, stressing that it is probably not conceivable to integrate polyphenols through the diet, but that these will have to be administered as concentrated extracts to reach the minimal useful concentration, and that doubts arise regarding the possible safety of such concentrations given the known possible adverse effects. A more comprehensive discussion of the potential risks and the side effects of high doses of polyphenols in mice and human models was instead added in the paragraph 8 (pag.16 lines 661-686).

  1. There are some minor grammatical errors that need to be corrected. Done

Reviewer 4 Report

This review by Pratelli et al. looks at a foodomics approach to IBD. This is an important and approach to this complex disease. The authors cite current literature. I do have some comments on the organization and specific points to make.

1. The omics chapters are too long. There is no need for two paragraphs consisting of more then 25 lines explaining the meaning of there relatively well known technologies. The authors should be more concise on the relevance to subject. 

2. Fig 1 explains a mechanism. The legend should be much more descriptive.

3. The case studies brought are interesting and relevant. It would be beneficial to the reader to have the authors sum up the different treatments of polyphenols to IBD in a table. This is common in reviews of this sort. It should list the compound, model (mouse, human, rat etc), if IBD was induced then how chemically  (DSS, TMBS) or genetically, the result and reference.

The paper is well written. A few 'track changes' slipped through the cracks and other small corrections.

1. Ln 199 after (67) comma is red

2. Ln 221 after (75) comma is red

3. Ln 411 period is red

4. Ln 421 the sentence starting with 'determining' is red

5. Ln 447 first person 'We' is inappropriate Change to 'humans'.

6. Ln 483 space between gut(184,185)

7. Ln 497 space between [187] and advanced

1.

Author Response

  1. The omics chapters are too long. There is no need for two paragraphs consisting of more then 25 lines explaining the meaning of their relatively well-known technologies. The authors should be more concise on the relevance to subject

We thank the reviewer for this comment. We modified the manuscript by joining the sections 6-9 in a single paragraph that discuss in a more concise manner the different omics approaches (genomic, transcriptomic, proteomic and metabolomic) that can be used to investigate the relationship between food and health.

  1. Fig 1 explains a mechanism. The legend should be much more descriptive.

We thank the reviewer for this comment. We modified the legend of Figure 1 to better describe the mechanisms induced by dietary polyphenols in IBD.

  1. The case studies brought are interesting and relevant. It would be beneficial to the reader to have the authors sum up the different treatments of polyphenols to IBD in a table. This is common in reviews of this sort. It should list the compound, model (mouse, human, rat etc), if IBD was induced then how chemically  (DSS, TMBS) or genetically, the result and reference.

We thank the Reviewer for this observation. We added a table (Table 1) that summarize the case studies concerning the use of polyphenols in IBD.

Minor

  1. Ln 199 after (67) comma is red. Done
  2. Ln 221 after (75) comma is red. Done
  3. Ln 411 period is red. Done
  4. Ln 421 the sentence starting with 'determining' is red. Done
  5. Ln 447 first person 'We' is inappropriate Change to 'humans'. Done
  6. Ln 483 space between gut (184,185). Done
  7. Ln 497 space between [187] and advanced. Done

Reviewer 5 Report

This manuscript as a pleasure to read. Suggest in abstract line 5 delete and . insert termed,

Author Response

This manuscript as a pleasure to read. Suggest in abstract line 5 delete and . insert termed,

We thank the Reviewer for the appreciation of the manuscript. We modified the abstract in accordance with the request.

Round 2

Reviewer 1 Report

To: International Journal of Molecular Sciences 

Dear EIC Prof. Dr. Maurizio Battino.

And dear section editor.

This is my review report for revised version of manuscript ID: ijms-2597656.

The manuscript was revised carefully. However, authors response to one main question which I asked in the previous report was not convinced me. Therefore, I decided to ask as a single question/comment again.

1.     What is difference and novelty of your study when compared by previously published review articles in the field, especially articles by Zhang et al. and He et al.? Please give more details. This is a major problem. The links for these studies are listed below.

https://www.frontiersin.org/articles/10.3389/fimmu.2021.671150/full

https://www.sciencedirect.com/science/article/abs/pii/S2214799321000709

Author Response

We thank the reviewer for this observation that give us the opportunity to clarify the aims of our review.  The beneficial effects of polyphenols in GI has been previously reported in other studies (Zhang et al. Front. Immunol., 2021) and He et al (Current Opinion in Food Science, 2021) . However, these studies focused their attention on protective effects of polyphenols in gastrointestinal tract highlighting the beneficial effects on microbiome.

Instead, our research explores the most recent evidence concerning the potential impact of polyphenols on individuals suffering from IBD, specifically focusing on Crohn's disease and ulcerative colitis. Our review goes deeper in the use of single polyphenols in IBD mouse and human models and the molecular pathways activated by them, emphasizing a multi-omics integration to clarify the relationship between treatment and host, treatment and microbiome, and the interactions between host and microbe playing a key role in IBD pathogenesis and response to polyphenols treatment.

We hope that our manuscript could be useful to provide new critical insight enabling precision and personalized nutrition approaches, potentially leading to beneficial outcomes and improvements in the prevention and treatment of IBD with polyphenols as supplement to conventional therapies, allowing to overcome their limitations such as their low bioavailability, low bio accessibility and side effects of which we have spoken specifically.

Round 3

Reviewer 1 Report

The authors should insert their response to my comment in the second version of revision in the Discussion section. This will gives clear view of comparison between this study and the previous ones.